# Automated Detection, Segmentation, and Classification of Pericardial Effusions on Chest CT Using a Deep Convolutional Neural Network

**DOI:** 10.3390/diagnostics12051045

**Published:** 2022-04-21

**Authors:** Adrian Jonathan Wilder-Smith, Shan Yang, Thomas Weikert, Jens Bremerich, Philip Haaf, Martin Segeroth, Lars C. Ebert, Alexander Sauter, Raphael Sexauer

**Affiliations:** 1Division of Research and Analytical Services, University Hospital Basel, 4031 Basel, Switzerland; adrianjonathan.wilder-smith@usb.ch (A.J.W.-S.); shan.yang@usb.ch (S.Y.); thomas.weikert@usb.ch (T.W.); martin.segeroth@usb.ch (M.S.); alexander.sauter@usb.ch (A.S.); 2Department of Radiology, University Hospital Basel, University of Basel, 4031 Basel, Switzerland; jens.bremerich@usb.ch; 3Department of Cardiology, University Hospital Basel, University of Basel, 4031 Basel, Switzerland; philip.haaf@usb.ch; 43D Center Zurich, Institute of Forensic Medicine, University of Zürich, 8057 Zürich, Switzerland; lars.ebert@irm.uzh.ch

**Keywords:** pericardial effusion, computed tomography, X-ray, AI (Artificial Intelligence), hemopericardium, DCNN (deep convolutional neural network)

## Abstract

Pericardial effusions (PEFs) are often missed on Computed Tomography (CT), which particularly affects the outcome of patients presenting with hemodynamic compromise. An automatic PEF detection, segmentation, and classification tool would expedite and improve CT based PEF diagnosis; 258 CTs with (206 with simple PEF, 52 with hemopericardium) and without PEF (each 134 with contrast, 124 non-enhanced) were identified using the radiology report (01/2016–01/2021). PEF were manually 3D-segmented. A deep convolutional neural network (nnU-Net) was trained on 316 cases and separately tested on the remaining 200 and 22 external post-mortem CTs. Inter-reader variability was tested on 40 CTs. PEF classification utilized the median Hounsfield unit from each prediction. The sensitivity and specificity for PEF detection was 97% (95% CI 91.48–99.38%) and 100.00% (95% CI 96.38–100.00%) and 89.74% and 83.61% for diagnosing hemopericardium (AUC 0.944, 95% CI 0.904–0.984). Model performance (Dice coefficient: 0.75 ± 0.01) was non-inferior to inter-reader (0.69 ± 0.02) and was unaffected by contrast administration nor alternative chest pathology (*p* > 0.05). External dataset testing yielded similar results. Our model reliably detects, segments, and classifies PEF on CT in a complex dataset, potentially serving as an alert tool whilst enhancing report quality. The model and corresponding datasets are publicly available.

## 1. Introduction

With the increasing use of Computer Tomography (CT) in medicine [1], pericardial effusions (PEFs) are often first diagnosed on CT [2]. A PEF is most commonly defined as a volume larger than 50 mL [2,3,4]. However, PEF diagnosis is often missed on CT as shown by Verdini et al. with a sensitivity of only 59.0% for pericardial disease [5]. Whilst volumetry is the most accurate method to diagnose PEF, taking into account the complex three-dimensional geometrical structure of the pericardial sac [4,6,7], counting voxels is time consuming and impractical in clinical practice [8]. Therefore, an automatically generated segmentation and volumetry tool would substantially improve the quality of CT-based PEF diagnosis.

Whilst echocardiography is the primary tool for PEF diagnosis [6], CT plays an ever-increasing role. This is because CT is a “catch all” investigation for patients presenting *in extremis*, which requires prompt, thorough and rapid diagnostic workup to identify life threatening pathologies and guide management. Evidently, missing a PEF diagnosis in this context must be avoided. Additionally, the echocardiographic diagnostic accuracy of PEFs is affected by the presence of clots, complex loculations, posterior PEFs and post-surgical changes, making CT the primary and most straightforward tool for the investigation of these complications [9,10].

The effectiveness of deep convolutional neural networks for the automated analysis of chest CTs has been proven [11,12,13,14]. However, to date, only two studies are available utilizing Artificial Intelligence to automatically identify or quantify PEF on CT [15,16]. Currently, the most commonly implemented deep learning architecture in medical imaging is the U-Net [17], which was adapted by Isensee et al. producing a robust tool called nnU-Net. nnU-Nets can automatically configure themselves, including preprocessing, network architecture, training and post-processing for any new task [18]. Its advantages have been demonstrated by winning multiple Medical Image Computing and Computer Assisted Interventions challenges.

Our model presents the first openly available dataset and tool to automatically segment, quantify and classify PEFs on CT. It fulfills the clinical need for a more reliable, more accurate, and faster diagnosis of PEF and hemopericardium, which performs reliably even in the presence of additional chest pathology. The incorporation of this tool into clinical practice may help reduce or even avoid missed diagnoses completely, as well as improve the time-to-diagnosis, ultimately improving patient outcomes.

## 2. Materials and Methods

The local ethics committee approved this retrospective study (Project ID: 2021-00946). For the external dataset consisting of anonymized post-mortem CT data, an ethical waiver was issued by the Ethical Committee of the Canton of Zurich (Project ID: 2022-00173). We structured our manuscript according to SPIRIT-AI and CLAIM [19,20].

### 2.1. Study Population

Two study cohorts were defined as follows. The positive cohort, consisting of simple PEF and hemopericardium cases, was identified through the search of structured radiology reports using the search terms “pericardial effusion” and “hemopericardium” from January 2016 to January 2021. No patient was present in both groups. The corresponding CTs including chest imaging were identified using the Radiology Information System/Picture Archiving and Communication system in a tertiary hospital. The search and the exclusion of patients without consent, duplicates and follow-up studies were carried out by Reader 2 (postgraduate year (PGY) 4). Additionally, a negative cohort was identified on a local database of CT studies without significant radiological findings by Reader 1 (PGY 2).

All CTs identified above were quality controlled by Reader 1 under the supervision of Reader 3 (Board certified, 15 years of cardiothoracic imaging experience). All hemopericardium cases were reviewed by Readers 1 and 2 to ensure an accurate diagnosis. Most cases had a concomitant aortic dissection, recent cardiac intervention, or chest trauma (*n* = 51/53, 96.23%). Studies with strong artifacts affecting the delineation of the heart, insufficient image quality, and extreme difficulty in delineating between PEF and pleural effusion were excluded with the recommendation from Reader 3. Studies in the PEF/hemopericardium and negative cohort were manually reviewed to ensure a PEF thickness of >4 mm and <4 mm, respectively [2], whilst blinded to the radiology report. In areas of diagnostic uncertainty, the consensus was achieved through discussion with Readers 2 and 3. A detailed overview of the patient selection is graphically presented in Figure 1. All data were anonymized.

### 2.2. Image Acquisition Parameters

CT scans were performed using four different models manufactured by Siemens Healthineers, Germany: Definition AS+ (*n* = 220), Definition Flash (*n* = 158), Definition Edge (*n* = 118), and Somatom Force (*n* = 20). Acquisition parameters were as follows: mean peak tube voltage 102.82 kVp (SD: 12.18), mean tube current time product 113.03 mAs (SD: 41.47), mean computed tomography dose index (CTDI) 4.07 mGy (SD: 3.62) and mean dose length product (DLP) 152.60 mGy*cm (SD: 162.07). A soft tissue kernel reconstruction (30f) of 1.0 mm served as the only input for the model.

Iopromide (Ultravist 370, Bayer Pharmaceuticals) was administered in 274 cases (arterial phase = 143, biphasic = 25, venous = 7, pulmonary arterial phase = 99).

### 2.3. Segmentation

All images with PEF were manually segmented by two readers Reader 1 (*n* = 187) and Reader 2 (*n* = 71) under supervision of Reader 3. Segmentation was standardized by using the same software and windowing. The three-dimensional segmented masks were used as a reference for training and testing.

To measure inter-rater variability, 40 studies initially segmented by Reader 2 were randomly selected and segmented by Reader 1; 20 with and 20 without contrast.

### 2.4. Model Training, Validation, and Testing

The model architecture is presented in Table 1. Hemopericardium studies were often not chest only CTs (*n* = 26). Therefore, pre-processing was carried out to crop the CT to the chest region, which is predicted by Hofmanninger et al.’s lung segmentation model [21].

The positive and negative cohorts (*n* = 516) were divided randomly into a training/validation (*n* = 316) and a test cohort (*n* = 200) whilst preserving the distribution of cases with and without contrast administration. Hemopericardium cases were divided with increased weighting towards the test cohort in order to ensure good statistical power of the accuracy of hemopericardium segmentation and classification. Therefore, hemopericardium compromised only 10% of the positive cohort within the training cohort (*n* = 14), matching our local hemopericardium prevalence amongst PEFs. The remaining were used in the test cohort (*n* = 38). A deep convolutional neural network using nnU-Net [18] was trained and validated with a 5-fold cross validation. Each fold took 1.5 days. 

### 2.5. Hardware and Software

The images were organized, viewed, labeled, predicted, pre- and post-processed on the web-based image platform NORA [https://www.nora-imaging.com/ (accessed on 15 February 2022)] which is installed on an inhouse server with GPU Tesla T4. The training was performed on an Ubuntu workstation with 12 Cores CPU, 64 GB RAM and two Nvidia RTX 2080Ti.

### 2.6. Classification of Hemopericardium

The model’s prediction mask was used to extract a Hounsfield unit (HU) for each voxel segmented. All values outside the range of 0 to 80 HU were removed as these are outside the HU range for fluid and blood. The remaining values were used to compute a median HU per case, as demonstrated in Figure 2. A total of 38 hemopericardium cases from the hemopericardium sub-cohort together with one incidentally identified hemopericardium case in our positive cohort were compared to the remaining 61 simple PEF cases (without hemopericardium).

### 2.7. The Model’s Output

The resulting output from the segmentation prediction will be: the presence of PEF > 50 mL, volume in mL, large PEF (>100 mL) and the HU. The presence of hemopericardium and a PEF volume > 100 mL could set off an alert for the emergency physicians and radiologists.

### 2.8. External Data Set

Our model was tested on an external dataset consisting of 22 post-mortem CTs (PMCTs) with autopsy-confirmed hemopericardium, performed on a Siemens Definition Flash CT scanner. The image acquisition parameters are based on the chest and abdomen scanning protocol described by Flach et al. [22]. Patient selection criteria are described by Ebert et al. [23]. The CTs were cropped using the aforementioned technique at our institution. Our model’s performance was compared to the external institution’s post-mortem volume measurements and manual segmentation masks, the methodology of which is described by Ebert et al. [15].

### 2.9. Statistical Analyses

To evaluate PEF detection and segmentation performance, we used sensitivity, specificity, and Dice coefficient. Volumetry was compared using Bland–Altman analysis and by plotting the reference volumes against the predicted volumes and calculating Pearson’s correlation coefficient (r^2^). To compare model performance within the sub-groups and to the inter-rater agreement, we used the Dice coefficient, *t*-test, Mann–Whitney U test, and intraclass correlation coefficient (ICC). The receiver operating characteristic (ROC) curve was used to identify a HU threshold for the diagnosis of hemopericardium. Dice coefficient was computed on local software. The remaining statistical tests were computed using the packages “pROC” and “irr” using RStudio (Rstudio, PBC, Boston, MA, USA) [24,25], and visualized graphically using “ggplot2” and “tidyverse” [26,27].

### 2.10. Publicly Available Data

To encourage the use of this model and ensure the validity of our results, the CT datasets, reference segmentations and model code are openly available online [28].

## 3. Results

### 3.1. Study Population

The training and test cohorts did not significantly differ regarding both age (*p* = 0.695) and reference volume (*p* = 0.492). The training dataset had a mean volume of 212.82 ± 220.23 mL (±standard deviation) compared to 198.20 ± 159.78 mL for the test dataset. There was a total of 258 reference segmentations. Using the definition of >50 mL for the diagnosis of PEF, there were a total of 217 PEFs; 128 in the training cohort and 89 in the test cohort. The patient characteristics within the test cohort are summarized in Table 2.

### 3.2. Inter-Reader Versus Reference-Prediction

The inter-reader dataset comprised 40 cases, and the following analyses are based on these cases only and presented in Table 3. The ICC of these segmentation volumes was 0.970 (95% CI 0.951 < ICC < 0.983). A violin plot of all segmented volumes and example segmentations is shown in Figure 3.

### 3.3. Model Performance and Effect of Confounding Factors

Hereon, the analyses are based on the positive test cohort only (*n* = 100), which are presented in Table 4. Appendix A presents an in-depth analysis. Figure 4a shows an example case with good quality model segmentation—unaffected by contrast administration. Figure 4b shows an example case with a concomitant pleural effusion. Here, the segmentation quality is reduced due to the artifact. Importantly, the pleural effusion was not segmented despite being immediately adjacent to the PEF.

### 3.4. Detection of Pericardial Effusion

A reference segmentation volume >50 mL was used as the true positive. True negative cases were manually identified, and additionally, included reference volumes <50 mL. The sensitivity and specificity were 97.00% (95% CI 91.48–99.38%) and 100.00% (95% CI 96.38–100.00%).

### 3.5. Hemopericardium Classification

The median HU was extracted from each CT study using the predicted segmentation mask in the positive test cohort (*n* = 100), of which 39 had hemopericardium. The results are presented in Table 5. A threshold HU value of 24.5 was identified to diagnose hemopericardium with an AUC of 0.944 (95% CI 0.904–0.984) and is demonstrated in Figure 5 alongside two further ROC curves consisting only of cases with or without contrast. Both Youden’s formula and “closest top left” formula yielded the same threshold result with a sensitivity and specificity of 89.74% and 83.61% (circled in red).

### 3.6. External Validation

PMCTs of deceased patients suffering from hemopericardium were used for external validation. The calculated volumes using autopsy, reference and model are shown in Table 6. The Dice coefficient comparing reference and model segmentations was 0.66 ± 0.14 (median = 0.67). A violin plot of all segmented volumes and example segmentations is shown in Figure 6. The mean HU extracted from these PMCTs with hemopericardium was 37.36 ± 5.01, which is similar to hemopericardium cases in our test cohort (*p* = 0.387) and significantly larger compared to non-hemopericardium PEF cases in our test cohort (*p* < 0.001). Using 24.5 HU as a cut-off value, all external cases were diagnosed with hemopericardium with a corresponding sensitivity of 100.00% (95% CI 84.56–100.00%).

## 4. Discussion

We developed and comprehensively tested a tool for the automatic detection, volumetry and classification of PEFs using a complex set of clinical cases, containing cases of simple PEF, hemopericardium and a negative control without PEF. A highly accurate detection (97% sensitivity and 100% specificity) and good segmentation accuracy (Dice 0.75) were achieved. Of note, the volume difference and Dice coefficient between the reference and model were better than those of the inter-reader. The model’s performance was not significantly affected by acquisition parameters, such as contrast administration nor by clinical pathologies, such as PEF volume, hemopericardium, pleural effusions, and other radiologically identified pathology. A robust method using HUs for the classification of hemopericardium was developed, which had a sensitivity and specificity of 89.7% and 83.6%, which was unaffected by contrast administration. HUs are highly effective for the classification of PEFs [29]. Using the median HU of an entire PEF segmentation is a simple and reproducible method, superior to the current approach of viewing HU values in randomly selected areas of the PEF (with corresponding inter and intra observer variability). To our knowledge, this is the only study to date where this methodology is carried out. However, a few false positive cases were found, in particular, two CTs with a contrast of patients with pericarditis, resulting in pericardial enhancement affecting the HU values. We encourage that HU should not be used in isolation, but in addition to radiological and clinical findings.

Similar to other recent state-of-the-art Convoluted Neural Network (CNN) models, we show great utility and performance despite a highly complex dataset [30,31,32]. In fact, the high diagnostic accuracy in this study is comparable to Ay and Kahraman’s study comparing the manual diagnostic accuracy of CT and echocardiography in PEF detection after open heart surgery [33]. CT had a much lower false negative risk compared to echocardiography (8% vs. 61%) in a complex patient database that may be comparable to ours. Liu et al. [16] and Ebert et al. [15] investigated the feasibility of CT-based PEF segmentation on limited case numbers (*n* = 25 and *n* = 28, respectively). They both used two-step approaches and did not publish their model publicly. Both studies trained their models slice by slice, in comparison to more appropriate three-dimensional training in our study. Additionally, neither of the studies offers a PEF classification tool. Liu et al. analyzed the segmentation performance of two different neural network architectures. The best performing model was U-Net, which forms the basis of the nnU-Net architecture utilized in our study. Their model’s performance was similar to ours (Dice 0.77) but may not be comparable as no negative control cohort was included in the training or testing cohort. Ebert et al. studied whether automatic segmentation of hemopericardium on PMCTs was feasible and had 25 negative control cases. Direct analysis of their model is not possible because a commercial software was used. Ebert et al.’s dataset was used to perform an external validation. Here, we found a good segmentation quality (Dice 0.65) but a large difference in volume segmentation. This is explained by the difference between PMCTs and CTs of living patients. First, the images are higher resolution and without contrast. Second, radiological findings are starkly different, for example, the heart chambers collapse, occasionally contain air and hematomas, and therefore, the hemopericardium is denser and more inhomogeneous. This and the fact that PMCTs were not part of the training dataset might explain the lower quality of the segmentation.

### Clinical Implications

This automatic identification, volumetry and classification tool can facilitate and improve radiology triage and workflows as well as report quality. A well embedded tool could directly alert emergency physicians to the presence of hemopericardium and/or large PEF, with its association with cardiac tamponade [4]. Early CT has previously been shown to facilitate rapid patient management in the emergency department [34], and our tool may further enhance it. Additionally, this tool can make PEF volume and HU measurements objective, replacing current semi-quantitative classifications plagued by variable cut-off thresholds and inter and intra-reader variability [10,35]. As long as input parameters are standardized, this automatic tool can standardize these measurements, and therefore, be used to identify a less arbitrary cutoff for distinguishing between PEF and physiological pericardial fluid, taking into account age, sex, height, and concomitant disease. This will allow more accurate detection of PEF and hemopericardium which may reduce unnecessary echocardiograms or identify patients where an echocardiogram is urgent. It may also allow better prognostication. The incorporation of this tool into clinical practice requires seamless communication between the Radiology Information System/Picture Archiving and Communication System and the model. In addition to this infrastructure, a simple output is required which should be easy and intuitive for the radiologist or clinician to access.

There are several limitations to our study. First, the model was trained on the latest generation CTs from a single scanner manufacturer at a single institution. While the use of older CT scanners could theoretically reduce the diagnostic accuracy, we at least showed a good performance on CTs from a different institution on a completely different patient group: post-mortem hemopericardium cases. This suggests that the model’s performance is highly robust in different protocols, institutions and patient groups. Second, the ground truth was identified using the radiology report, which was mitigated by the use of structured reports in our institution. The two-dimensional CT diagnosis of PEF was used in our inclusion criteria, which resulted in a total of 29/258 (11.24%) PEF cases >4 mm which turned out to be physiological pericardial fluid volumes (<50 mL). No alternative mechanism exists to circumvent this drawback whilst maintaining a large case number. Third, inter-reader variability was worse compared to reference-prediction variability. This may be because segmentation style and errors have a proportionately large effect on outcome in complex, narrow and often small structures, such as PEFs.

## 5. Conclusions

This study trained and tested an automated tool using nnU-Net for the detection and segmentation of PEFs and additionally used HUs to classify PEF as hemopericardium or simple PEF. This tool is highly accurate for the automatic detection, volumetry, and classification of PEFs, even surpassing human inter-reader variability. Since this tool was trained using a complex dataset, it can be easily incorporated into clinical practice.

## Figures and Tables

**Figure 1 diagnostics-12-01045-f001:**
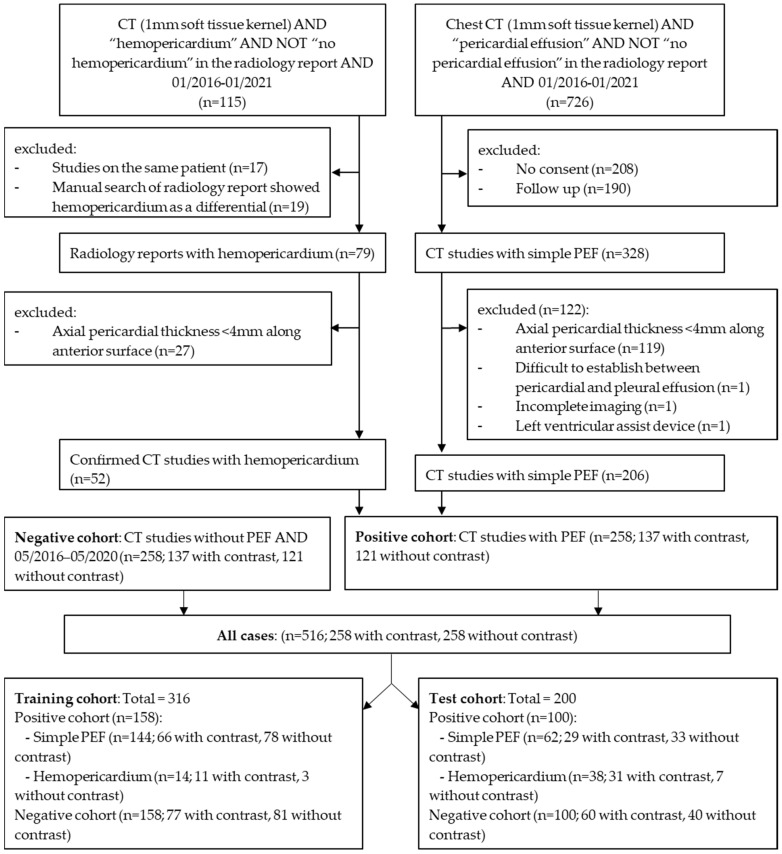
Patient selection flow chart, demonstrating the inclusion and exclusion criteria as well as numbers of studies with or without contrast. (PEF: pericardial effusion). The external dataset is not included in the flowchart.

**Figure 2 diagnostics-12-01045-f002:**
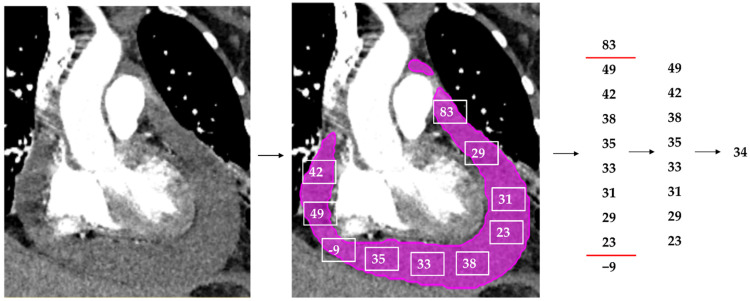
Simplified overview of the median Hounsfield unit (HU) extraction from an example CT with hemopericardium. A CT study with contrast depicting hemopericardium is segmented by the model (pink). Every voxel within the prediction mask generates a HU, visually depicted as the white squares with numbers within. All HUs outside of 0 to 80 are removed. The median HU of this PEF is selected using the remaining HUs and is the output (in this example 34 HU).

**Figure 3 diagnostics-12-01045-f003:**
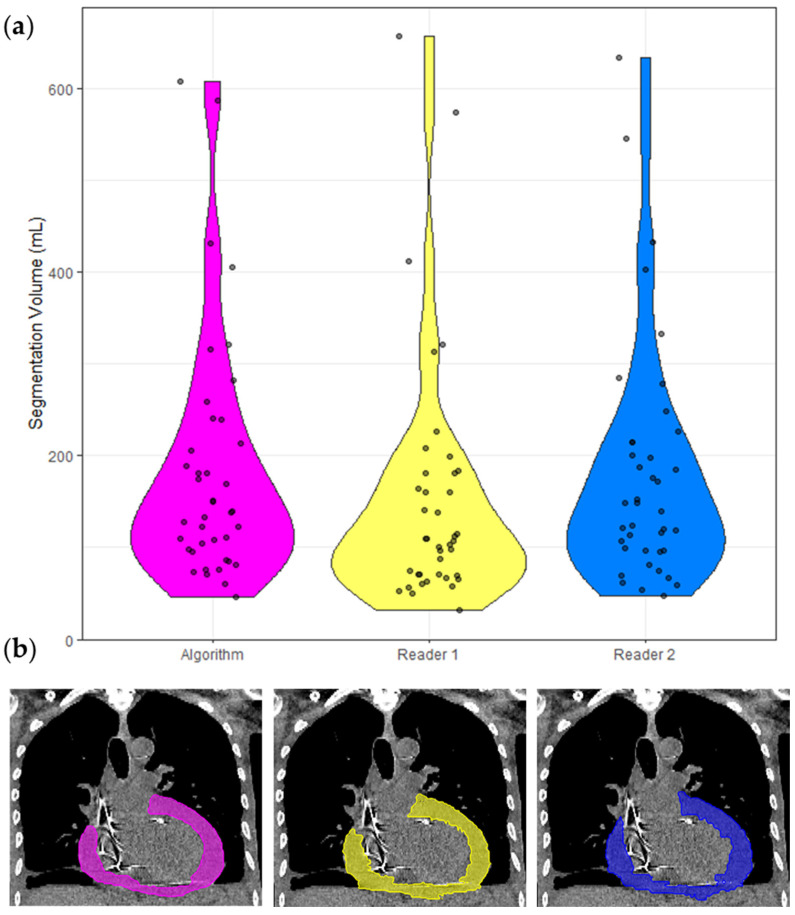
Segmentation volumes and a corresponding example CT with segmentation masks. (**a**) Violin plots of the segmentation volumes of the 40 cases in the inter-reader dataset. Each violin is color coded to match the corresponding segmentation in (**b**). Pink = model, yellow = Reader 1, blue = Reader 2. (**b**) Non-contrast CT chest of a 65-year-old shown with soft tissue windowing. Reader 1 segmented 183 mL, Reader 2 segmented 201 mL, and the model segmented 192 mL.

**Figure 4 diagnostics-12-01045-f004:**
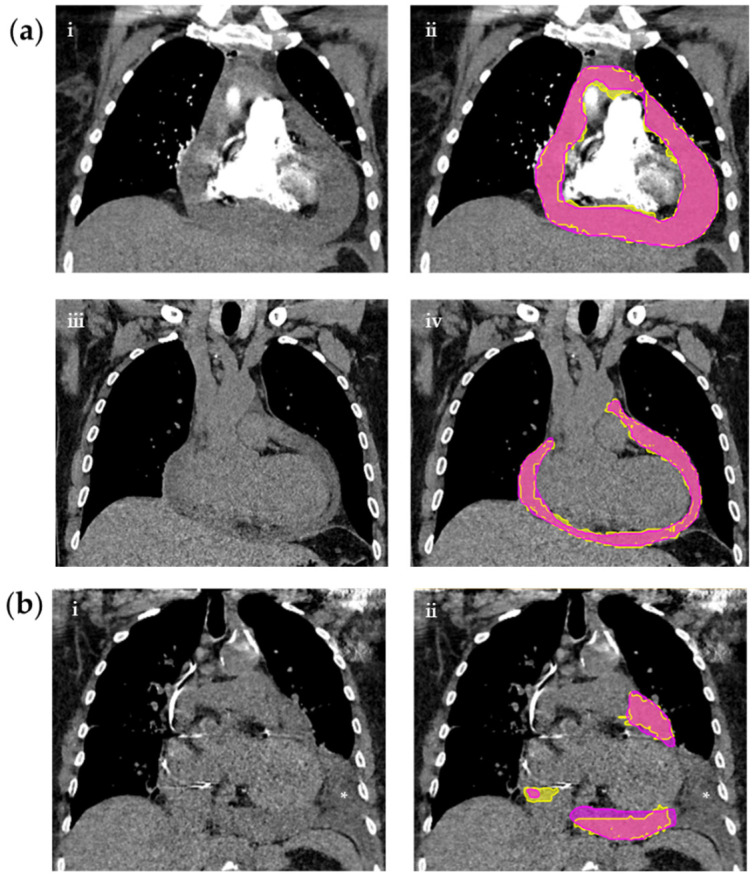
CT chests on the left with corresponding segmentations on the right of three different patients. (**a**) CT studies with or without contrast. (i) shows a contrast enhanced CT with hemopericardium without segmentation masks and (ii) with reference (yellow) and predicted (pink) masks. (iii) shows a non-contrast CT with PEF without masks and (iv) with reference (yellow) and predicted (pink) masks. (**b**) Non-contrast CT study with PEF and adjacent pleural effusion (*) (i) without masks and (ii) with reference (yellow) and predicted (pink) masks.

**Figure 5 diagnostics-12-01045-f005:**
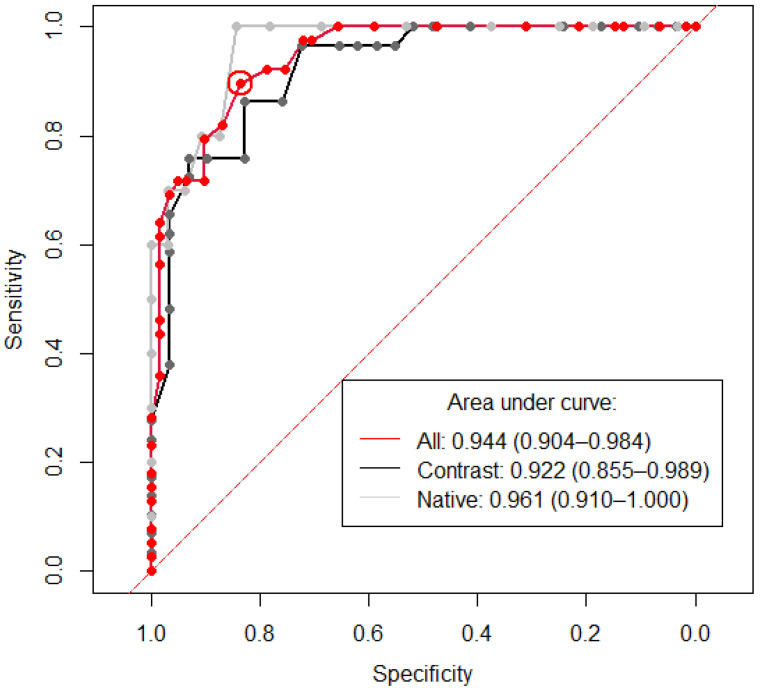
Receiver operating curve showing the sensitivity and specificity of Hounsfield units (HU) for the diagnosis of hemopericardium. The area under curve (AUC) is 0.944 (95% CI 0.904–0.984) for the whole positive cohort (*n* = 100) (red line). A threshold of 24.5 was identified, which is encircled in red and shows the value for the calculation of both Youden’s formula and the “closest top left” formula. The x and y values of this datapoint are 0.897 (specificity) and 0.836 (sensitivity), respectively. Labeled in black are all positive cases with contrast (*n* = 58). Here, a threshold of 26.5 was calculated using Youden’s formula, with a sensitivity and specificity of 0.862 and 0.828 respectively. Labeled in gray are all positive cases without contrast (*n* = 42). A threshold of 21 was calculated using Youden’s formula, with a sensitivity and specificity of 1.00 and 0.844, respectively. The dashed red line demonstrates the location of an AUC of 0.5, which is a random classifier and would, therefore, be of no diagnostic use.

**Figure 6 diagnostics-12-01045-f006:**
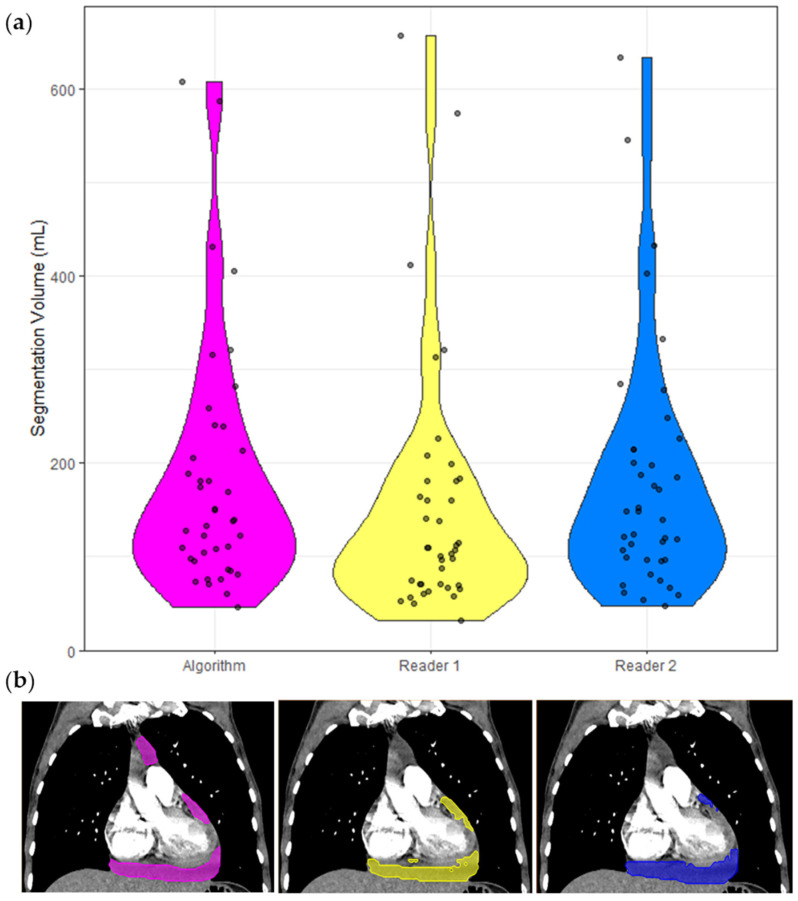
Segmentation volumes and a corresponding example CT with segmentation masks. (**a**) Violin plots of the volumes of the 22 cases in the external dataset. Each violin plot is color coded to match the corresponding volume calculation in (**b**). Pink = Model, gray/no mask = Autopsy, yellow = Reference. (**b**) PMCT in soft tissue windowing of a hemopericardium with aortic dissection. The model segmented 257 mL, the volume extracted at autopsy was 350 mL and the Reference segmented 290 mL.

**Table 1 diagnostics-12-01045-t001:** Model architecture.

Design	Parameters
Image preprocessing	Image downsampling to 1 × 1 × 1 mm^3^
Hard-/Software	Matlab R2018b and Python 3.7 on a workstation with a consumer-grade graphic processor unit (Nvidia RTX 2080Ti).
Optimizer	SGD with Nesterov momentum (µ = 0.99)
Learning rate	Poly-learning rate schedule (initial 0.01)
Data augmentation	Gaussian noise and blur, brightness, contrast, simulation of low resolution, gamma correction and mirroring
Loss function	Dice and cross-entropy
Training procedure	1000 epochs × 250 minibatches, foreground oversampling
Inference procedure	Sliding window with half-patch size overlap. Gaussian patch center weighting
Architecture template	Encoder-decoder with skip connection, instance normalization, leaky ReLU, deep supervision
Intensity normalization	Global dataset percentile clipping, z-score with global foreground mean and s.d.
Image resampling strategy	In-plane with third-order spline, out- of-plan with nearest neighbor
Annotation resampling strategy	Nearest neighbor interpolation to original spatial resolution
Image target spacing	Lowest resolution axis tenth percentile
Patch size	(128, 128, 128)
Batch size	2
Ensemble selection	3D U-Net according to cross-validation performance

**Table 2 diagnostics-12-01045-t002:** Overview of patient characteristics in the test cohort.

	Positive Cohort	Negative Cohort	Significance
Mean age	63.1	55.9	*p* < 0.001
Sex (male)	*n* = 66	*n* = 57	*p* = 0.193

**Table 3 diagnostics-12-01045-t003:** Overview of inter-reader and reference-prediction results.

	Inter-Reader	Reference-Prediction
**PEF volume**	Reader 1: 152.92 ± 134.85 mL	Reader 1: 152.92 ± 134.85 mL
	Reader 2: 181.04 ± 131.55 mL	Prediction: 184.20 ± 132.19 mL
	(*p* = 0.105)	(*p* = 0.062)
**Correlation coefficient**	r^2^ = 0.930	r^2^ = 0.925
**Dice coefficient**	0.69 ± 0.10, median = 0.69	0.73 ± 0.10, median = 0.74

**Table 4 diagnostics-12-01045-t004:** Model performance and the effect of confounding factors.

	Reference-Prediction	Significance
**PEF volume**	Reference: 198.20 ± 159.78 mL	*p* = 0.875
	Prediction: 201.50 ± 160.93	
**Correlation coefficient**	r^2^ = 0.929	N/A
**Dice coefficient**	0.75 ± 0.01, median = 0.76	N/A
**Effect of confounding factors assessed with Dice coefficient:**		
Contrast administration	Contrast: 0.74 ± 0.19, median = 0.77	*p* = 0.908
	No contrast: 0.75 ± 0.09, median = 0.76	
PEF size (>50 mL vs. >50 mL)	<50 mL: 0.67 ± 0.17, median = 0.74	*p* = 0.086
	>50 mL: 0.75 ± 0.10, median = 0.77	
Hemopericardium presence	Hemopericardium: 0.72 ± 0.13, median = 0.73	*p* = 0.097
	No hemopericardium: 0.76 ± 0.09, median = 0.78	
Pleural effusion presence	Pleural effusion: 0.73 ± 0.12, median = 0.74	*p* = 0.116
	No pleural effusion: 0.77 ± 0.08, median = 0.79	
Other radiologically identified chest pathology	Visible pathology: 0.73 ± 0.12, median = 0.74	*p* = 0.061
	No visible pathology: 0.77 ± 0.09, median = 0.78	

Radiologically identified pathology was defined as radiological evidence of chest trauma, infection, tumor, and post-operative changes.

**Table 5 diagnostics-12-01045-t005:** Using Hounsfield units for the classification of hemopericardium or simple PEF, and the effect of contrast on the classification.

	Hemopericardium	Simple PEF	Significance
**Hounsfield unit (HU)**	36.10 ± 9.72, median = 36	19.20 ± 5.52, median = 18	*p* < 0.001
**Affect of contrast on Hounsfield unit:**			
Presence of contrast	36.86 ± 9.65, median = 36	20.97 ± 6.35, median = 19	*p* < 0.001
No contrast	33.99 ± 10.08, median = 35	17.59 ± 4.12, median = 17	*p* < 0.001

**Table 6 diagnostics-12-01045-t006:** Volume of pericardial effusion identified on CT using autopsy, reference, and model.

	Autopsy-Reference	Reference-Prediction	Autopsy-Prediction
**PEF volume**	Autopsy: 488.33 ± 232.41 mL	Reference: 515.26 ± 216.67 mL	Autopsy: 488.33 ± 232.41 mL
	Reference: 515.26 ± 216.67 mL	Prediction: 301.69 ± 118.92 mL	Prediction: 301.69 ± 118.92 mL
	(*p* = 0.283)	(*p* < 0.001)	(*p* = 0.019)

## Data Availability

Algorithm code as well as corresponding datasets are openly available at https://doi.org/10.5281/zenodo.6384747 (accessed on 1 April 2022) [28].

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
