# Peer review of "Automated Detection, Segmentation, and Classification of Pericardial Effusions on Chest CT Using a Deep Convolutional Neural Network"

_diagnostics, 2022, doi:10.3390/diagnostics12051045_

Round 1

Reviewer 1 Report

The study nicely validated a CT software based on deep convolutional networks for the for detection and automated analysis of pericardial effusion. The aim was to support reporting activity with new tools, the study was well conducted from a methodological point of view.

Patients selection and type and sample size are carefully chosen to achieve robust results.

The are some point to improve to ease readers' understanding:

  1. Introduction should be shorter and more focused on CT.
  2. A table with patients’ characteristics should improve general comprehension (paragraph 2.6 could be summarized in a table).
  3. All dataset were achieved with latest generation scanners and you have not excluded any datasets for motion or heartbeat artifacts. Do you think using older scanners (below 128 slices) could affect the accuracy of the results?

Figures and plots are clear and well described.

Discussion is well written but it’s too long and it’s a bit difficult to follow, you can focus on the most important previous paper (in my opinion no more than two).

Bibliography is updated.

Reviewer 2 Report

In this paper, the authors discussed a tool for PEF identification. The authors also presented some results. The manuscript has several issues. The presentation is not good. Difficult to read. Plz, address the following comments.

  1. The abstract should first discuss the motivation for developing the tool for PEF identification.
  2. The authors should highlight the contributions at the end of the introduction. What’s novelty?
  3. Please create a paragraph to discuss the structure of the manuscript.
  4. Where is the model architecture?
  5. The authors discuss the tool architecture.
  6. The results are not well presented. Need to show results either in tabular form or using standard representations.
  7. No comparison is provided with the existing works [15,16]
  8. What is the data source?
  9. Don’t see any real contributions
  10. The keyword mentioned “deep learning”. I don’t see any deep learning.
  11. No discussion of related works

Reviewer 3 Report

This work uses existing deep learning techniques to automatically identify and segment Computed Tomography (CT) images to achieve PEF diagnosis. However, the U-Net used in this paper is a well-known model in the field of medical imaging. Furthermore, this paper does not present a new processing method for chest CT image. Thus, from an AI or deep learning perspective, nothing in this paper is substantially innovative. The authors should review some recent studies that contain recent developments in deep CNN, e.g., Classification of Diabetic Retinopathy from OCT Images using Deep Convolutional Neural Network with BiLSTM and SVM; Learning a deep predictive coding network for a semi-supervised 3D-hand pose estimation; Dynamic hand gesture recognition based on short-term sampling neural networks; and A Smartphone-Enabled Fall Detection Framework for Elderly People in Connected Home Healthcare.

The collection of chest CT images is the authors' contribution. However, these datasets are too tiny (only 316 cases), resulting in unreliable test results on CNN models. Furthermore, rather than sharing datasets only during review, authors should permanently publish the datasets they collect for future research. Finally, the authors should provide additional details on these datasets.

Round 2

Reviewer 2 Report

Thank you for addressing the comments. Good luck!